# HOW TO EVALUATE MONOCULAR DEPTH ESTIMATION?

## ABSTRACT

Monocular depth estimation is an important task with rapid progress, but how to evaluate it remains an open question, as evidenced by a lack of standardization in existing literature and an unhelpfully large selection of evaluation metrics whose trade-offs and behaviors are not well understood. This paper contributes a novel, quantitative analysis of existing metrics in terms of their sensitivity to various types of perturbations of ground truth, emphasizing comparison to human judgment. Our analysis reveals that existing metrics are severely under-sensitive to curvature perturbation such as making flat surfaces wavy. To remedy this, we introduce a new metric based on relative surface normals, along with new depth visualization tools and a principled method to create composite metrics with better human alignment. All code, data, and tools will be open-sourced.

## 1 INTRODUCTION

Monocular depth estimation (MDE) is the task of estimating pixelwise depth from a single RGB image. It has become a standard task with rapid progress, due to its importance for many applications, such as robotics, AR/VR, and the generation of images, videos or 3D worlds.

However, how to evaluate monocular depth estimation remains an open problem. This is reflected by a lack of standardization in existing literature, and a bewilderingly large menu of evaluation metrics. It is in fact possible for two papers (e.g., (Bochkovskii et al., 2025; Wang et al., 2025a)) to report 5 to 7 different metrics each but with zero overlap[1].

Even when models are compared under the same set of metrics, it is often not clear how to interpret the results. If a model performs well under one metric (say AbsRel after affine alignment on disparity) but poorly under another (say $\delta^1$ with no alignment), what does it mean? Often there lack clear answers, because the trade-offs between the different metrics are not well understood.

The unusually large selection of metrics arises from a combination of decisions that need to be made when designing an evaluation metric. For example, the decisions can include how to compute error of individual depth values against ground truth (by difference in depth, log depth, or inverse depth), whether and how to discount error for faraway points, whether to binarize the error and if so with what threshold, and how to account for the unknown global scale through alignment (in depth or inverse depth), and whether to also allow an unknown global shift.

The complexity of these decisions is due to several issues inherent to monocular depth estimation that preclude straightforward comparisons of individual depth values. One is scale ambiguity. The global scale of a scene is fundamentally ambiguous: a scene can be a miniature replica of a larger identical scene. This scale ambiguity can also occur locally in a scene: it can be impossible to tell from a single image whether an airplane in the sky is a big one far away or small one close by. Even without any scale ambiguity, another issue is the unbounded range of possible depth values in the same scene—consider an ocean that extends to the horizon. In such cases, some form of normalization or weighting is necessary to prevent errors on faraway objects from dominating.

To make matters worse, the predicted depth values alone do not reconstruct a 3D shape because it does not tell us the X and Y coordinates in 3D; for that we also need the camera intrinsics, which are often unknown and need to be predicted. But camera intrinsics can be ambiguous from a single image and the prediction can be reasonable but off. And the predicted depth and predicted camera intrinsics can together give the a reasonable 3D shape but have large errors when evaluated separately.

---

[1]Two metrics can share the same name but are different due to differences in depth alignment.

In this paper we seek to improve our understanding of how to evaluate monocular depth estimation. Our goal is to study various evaluation metrics and shed light on their trade-offs and behaviors, and to develop a principled method to customize or combine metrics based on the preferences of downstream applications. It is worth noting that we do not aim to propose a new metric that supersedes all existing ones, recognizing that the best metric can depend on the specific downstream application.

**Our Approach:** Our main approach is to quantify the sensitivity of each metric to various types of perturbations of the ground truth depth. In particular, we focus on 6 types of perturbations—surface orientation, camera intrinsics, relative scale, curvature, affine transform, and boundary—which are interpretable and representative changes to ground truth depth that could shed light on the behavior of a metric. The definition of each perturbation is given in Section 3.

Given these perturbations, we can measure the sensitivity of a metric to each perturbation, and compare the sensitivity between metrics. The basic idea is that for each perturbation, we can establish an exchange rate between two metrics under this perturbation. For example, to represent the same surface orientation perturbation, we need 2 units of metric A but 1 unit of metric B. This is an exchange rate of 2:1. We can then compare exchange rates across metrics and perturbations to better understand the behavior of various metrics in terms of how they respond to different perturbations.

Note that the exchange rates must be interpreted in comparison, not in isolation. An exchange rate of 20:1 between metric A and B under a perturbation (say surface orientation) does not tell which metric is more sensitive because the unit of each metric can be chosen arbitrarily (e.g., meter instead of millimeter). But if we discover an exchange rate of 2:1 between the same two metrics using the same units under another perturbation (say relative scale), we can conclude that compared to metric B, metric A is *more* sensitive to surface orientation than to relative scale, because 1 unit worth of surface orientation perturbation under metric B translates to 20 units of metric A, whereas 1 unit worth of relative scale perturbation under metric B only translates to 2 units of metric A. In other words, relative to metric B, metric A amplifies the surface orientation perturbation.

**Human Sensitivity:** We also apply our sensitivity analysis to human judgment. Human judgment can serve as a useful reference for sensitivity comparisons because (1) many generative applications produce content for human consumption and (2) the human visual system remains the best general-purpose depth perception system and human judgment may provide clues on what is and is not important for achieving human-level visual capabilities.

We measure the sensitivity of human judgment to various perturbations. Using a collection of synthetic test scenes and varying amounts of perturbations, we ask human annotators to judge whether a (possibly) perturbed depth map is the ground truth of a given RGB image. We then average the binary annotations to estimate the exchange rates of human judgment versus other metrics.

To help human annotators examine the reconstructed geometry, we introduce two new visualization tools: Textureless Relighting and Projected Contours. These new tools overcome the limitations of conventional visualizations such as textured point clouds, which can mask geometric artifacts because, like in video games, texture maps can create fake perceived geometry. Beyond measuring human sensitivity, the tools can also be useful in developing future depth models.

Using human judgment as a reference yields interesting findings about existing metrics. Humans are sensitive to affine transforms of depth or disparity, but many metrics perform affine alignment and are thus completely insensitive. Most notably, all widely used metrics have very poor sensitivity to curvature perturbation (e.g., making a flat plane wavy).

**Sensitivity Aligned Composition:** In addition to improving our understanding, our sensitivity analysis enables us to combine existing metrics to align with a specific sensitivity profile, such as that of humans or a downstream application. The basic idea, which we call "sensitivity aligned composition (SAC)", is to combine existing metrics through some parametric form of composition such as a weighted average, and optimize the composition parameters such that the combined metric achieves the designed sensitivities to a given set of perturbations.

Because existing metrics are overly insensitive to curvature perturbation compared to human judgment, we propose a new metric RelNormal, which is based on relative surface normals and thus sensitive to curvature perturbation. Combining RelNormal and a subset of existing metrics, we propose SAWA-H (Sensitivity Aligned Weighted Average based on Human judgment), a new, composite metric that aligns better with human judgment than all existing metrics.

**Summary of Contributions:** Our contributions are three fold. (1) We conduct a novel, quantitative sensitivity analysis of the commonly used metrics under various perturbations. (2) We introduce two new tools of depth visualizations and measure sensitivity of human judgment, revealing that existing metrics are overly insensitive to curvature perturbation. (3) We introduce SAWA-H, a new metric that better aligns with human judgment via optimized combination of a set of base metrics; we also introduce RelNormal, a new base metric designed to enable better human alignment. All code, data, visualization tools will be open sourced.

## 2 RELATED WORK

**Monocular Depth Evaluation Metrics.** Conventional metrics directly evaluate the difference between predicted and ground truth depth. They evaluate accuracy of ordinal relationship (Zoran et al., 2015; Chen et al., 2016) or per-pixel depth difference (Eigen et al., 2014; Saxena et al., 2008). Recently, Koch et al. (2018); Chen et al. (2019); Örnek et al. (2022); Talker et al. (2024); Pham et al. (2024) noted that these metrics are not sensitive to over-smooth boundaries and error at predicting planes, and new metrics were proposed to evaluate boundary sharpness (Koch et al., 2018; Chen et al., 2019; Bochkovskii et al., 2025) or difference in 3D (Koch et al., 2018; Örnek et al., 2022; Wang et al., 2025a). Though prior work noted that existing metrics are not sensitive to some errors, there lacks a principled way to systematically study behavior of different metrics. We attempt to fill this gap.

**Monocular Depth Estimation Methods.** There is rapid development in MDE in recent years. It is worth noting that a decent number of methods predict affine-invariant depth or disparity (Birkl et al., 2023; Yang et al., 2024a;b; Ke et al., 2024; Fu et al., 2024), and during evaluation, they perform affine alignment on depth or disparity before computing metrics. Li et al. (2025) pointed out two issues with alignment: unfair comparison between methods that use different alignment, and alignment is sensitive to outliers. In our analysis we further find that alignment methods are highly sensitive to errors that humans judge to be minor.

**Human Sensitivity and Visualization Tools** Though human vision is known to be accurate and fairly robust, humans still exhibit peculiarities when judging 3D structure. When constructing the OASIS dataset, Chen et al. (2020) observed that annotators judged the shape correctly (i.e. the relative normals) but often made mistakes when estimating the overall orientation. Linsley et al. (2025) find that though humans exhibit similar capabilities as deep neural networks when estimating depth order, humans are much better at visual perspective taking (answering "can object A see object B?"). This suggests that standard 3D geometric evaluations fail to capture the aspects of a scene that humans are sensitive to.

When visualizing depth, monocular depth estimation works typically plot heatmaps of either depth or disparity (Birkl et al. (2023); Eigen et al. (2014); Zoran et al. (2015)). Other works also display novel views of the unprojected depth map with the points colored by the corresponding pixel in the original image Wang et al. (2025a); Yin et al. (2021). Notably, the shading in these point clouds does not correspond with the true geometry. Wang et al. (2025a), Cao et al. (2022), and others additionally display a gray mesh object with shadows and specular highlights rendered from the original view. This is significant, as Liu & Todd (2004) finds that shadows, specular highlights, and other aspects of normal shading greatly improve humans' ability to discern curvature. We further expand on these visualization tools in section 4.1.

## 3 SENSITIVITY ANALYSIS

**Perturbations:** We study sensitivity of metrics to a set of interpretable perturbations.

- **Surface orientation perturbation** refers to perturbing depth so that surface normal of unprojected 3D geometry equals rotating ground truth surface normal by the same amount. Fig. 1 shows an example.
- **Camera intrinsics perturbation** refers to perturbing both focal length and depth, where unprojecting perturbed depth to 3D using perturbed focal length gives almost perfect shape. Fig. 1 shows an example. Note that despite 3D geometry is similar, difference between predicted depth and ground truth is huge (Fig. 1 right).
- **Relative scale perturbation** refers to perturbing relative scale between near and far pixels, e.g. foreground and background objects.

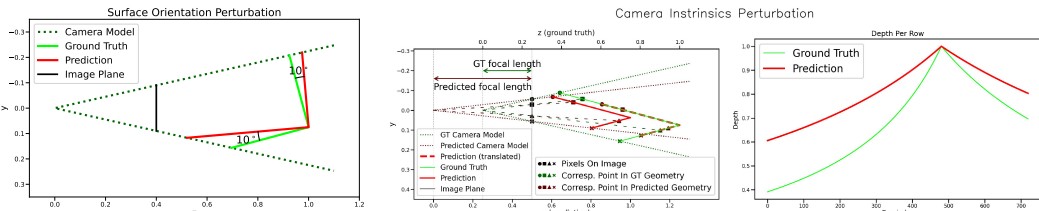

Figure 1: (a) Surface orientation perturbation (side view). Orientation of prediction (red) equals rotating that of ground truth (green) around x-axis by $-10°$. In side views through the paper, we show simple geometry whose depth is the same on each row. (b) Camera intrinsics perturbation. Left (side view): prediction is made under wrong focal length (2x ground truth focal length), while predicted geometry (red) has shape similar to ground truth (green). Predicted geometry can overlap well with ground truth through translation (red dashed). Right: predicted depth substantially differs from ground truth depth.

- **Curvature perturbation** refers to perturbing curvature of un-projected 3D geometry, e.g. making a smooth plane as bumpy. We consider curvature perturbation under two frequencies. Please refer Appendix for details.
- **Affine transform perturbation** refers to perturbing ground truth depth by making affine transform of depth or disparity.
- **Boundary perturbation** refers to perturbing depth of pixels near occlusion boundaries to be floating in the air.

Please refer to Appendix for details of perturbation algorithms.

**Exchange rates:** For each metric and perturbation, we assess the response on metric values when applying that perturbation to ground truth. Then we compare sensitivity of different metrics across different perturbation types to examine which metric is more or less sensitive to which perturbation. To facilitate such comparison, we introduce the notion of exchange rate.

The exchange rate between metrics A and B under perturbation P, $R(A; B|P)$ is (informally) defined as the ratio of changes of metrics A and B when perturbing the ground truth depth by the same intensity of perturbation P. Without loss of generality, we assume all metrics can be standardized into a form that starts with zero for ground truth depth and increase as the predicted depth deviate from ground truth.

For example, if after the same perturbation, the value of metric A (say RMSE) increases from 0 to 2 and value of metric B (say AbsRel) increases from 0 to 0.5, the exchange rate between metrics A and B is $2/0.5 = 4$. This rate can be interpreted as: to represent the same intensity of perturbation P, 4 units of metric A are needed, while we only need 1 unit of metric B.

This intuitive notion of exchange rate is well defined if both metric A and B respond linearly to the intensity of the perturbation P. However, this is almost never the case. For example, some metrics are bounded and will eventually saturate for large perturbations. Therefore our formal definition of the exchange ratio needs to also handle nonlinear functions.

Formally, we treat metrics $A(x)$ and $B(x)$ as smooth functions of intensity $x$ of perturbation P, and define the exchange rate $R(A; B|P)$ to be the ratio between their derivatives at zero:

$$R(A; B|P) \triangleq A'(0)/B'(0). \tag{1}$$

In the words, we approximate each metric as a linear function of perturbation intensity in a small neighborhood around zero. This approximation is justifiable because as models improve, the neighborhood around zero becomes more important.

In practice, we approximate each metric through a least squares fit of a quadratic function to $ax^2 + bx$, and take the derivative of the quadratic function at zero. Using a quadratic form can better approximate plateauing metrics and removes the need to manually select the size of the neighborhood around zero.

| | Surf. Ori. | Cam. Intr. | Rel. Sc. | Curv., high freq. | Curv., low freq. | Af. Depth | Af. Disparity | Boundary |
|---|---|---|---|---|---|---|---|---|
| Boundary F1-No Align. | 0.03 | 0.04 | 0.03 | 1.62 | 0.79 | 0.03 | 0.04 | 2.18 |
| AbsRel-Disparity Af. | 0.60 | 1.62 | 0.09 | 0.23 | 0.23 | 2.20 | 0.00 | 0.29 |
| AbsRel-No Align. | 1.00 | 1.00 | 1.00 | 1.00 | 1.00 | 1.00 | 1.00 | 1.00 |

Figure 2: Exchange rate between each metric and AbsRel with no alignment. Because absolute scale of exchange rate is not important, for better visualization, each row is normalized. Perturbations from left to right: surface orientation, camera intrinsics, relative scale, curvature (high frequency), curvature (low frequency), affine transform of depth, affine transform of disparity, and boundary.

**Comparing Sensitivity:** It may appear that metric A is more sensitive than metric B to perturbation P if the exchange rate $R(A; B|P)$ is large, as metric A has a larger response. But this impression is false because units of metrics A and B can be arbitrarily chosen. For example, metric B and metric A can be identical except that A is measured in meters and B is in millimeters. Then the exchange rate is 1000, but metrics A and B are effectively identical.

This means that a single exchange rate $R(A; B|P)$ between A and B under perturbation P is not meaningful in isolation and must be interpreted in comparison with another exchange rate. Specifically, let $R(A; B|Q)$ be the exchange rate of the same metrics A and B under a new perturbation Q, using the same units of metric A and B. If the exchange rate under $P$ (for example 1000:1) is bigger than that under $Q$ (for example 2:1), then we can conclude that relative to B, metric A is more sensitive to perturbation P than Q. This is because 1 unit worth of perturbation P measured under metric B translates to 1000 units under metric A, whereas 1 unit worth of perturbation Q measured under metric B translates to only 2 units under metric A. In other words, compared to metric B, perturbation P causes bigger numerical changes of metric A than perturbation Q.

**Dataset for measuring sensitivity.** To estimate the exchange rate in practice, we need a dataset of 3D scenes with ground truth depth maps. This is because the sensitivity of a metric is often scene dependent, in which case we obtain the expected sensitivity averaged over multiple scenes.

We create a dataset of synthetic scenes using Infinigen Raistrick et al. (2023; 2024), a procedural generator of photorealistic nature and indoor scenes. We choose to use synthetic data because real-world data are limited in the diversity of scenes and availability of dense depth ground truth. For example, no pixelwise real-world ground truth is available for large natural scenes. We choose Infinigen because it covers both indoor and natural scenes and is easy to customize.

The dataset consists of 95 scenes. Among them, 56 are indoor scenes, which have short depth range and more regular shape, and 39 are nature scenes, which have long depth range and less regular shape. For each perturbation type, we perturb by at least 6 different intensities, with a total of 5320 perturbed depth in the dataset. Example scenes are included in Appendix E.

**An Example of Comparison of Existing metrics** Here we show an example of using our methodology to compare sensitivity between 3 metrics: AbsRel-No Align (AbsRel (Saxena et al., 2008) with no alignment), AbsRel-Disparity Af (AbsRel after affine alignment of disparity), and Boundary F1-No Align (Boundary F1 (Bochkovskii et al., 2025) with no alignment).

Fig. 2 shows the exchange rate with respect to AbsRel-No Align. Comparing Boundary F1-No Align and AbsRel-No Align, the exchange rate under boundary perturbation is the highest. This is expected as Boundary F1 is designed to capture boundary sharpness which is not reflected in AbsRel and other metrics. Comparing AbsRel-Disparity Af and AbsRel-No Align, exchange rate under affine transform of disparity is the lowest. This is also expected as affine alignment on disparity can perfectly align depth of this perturbation to ground truth.

## 4    HUMAN SENSITIVITY

We are particularly interested in how the sensitivity of existing metrics compares to human judgment. This is a useful question because many generative AI applications generate content for human consumption, and for such applications human judgment may be the only evaluation that matters. In addition, findings of human sensitivity can reveal what is and is not important for achieving human-level visual capabilities, which would be sufficient for robotic applications.

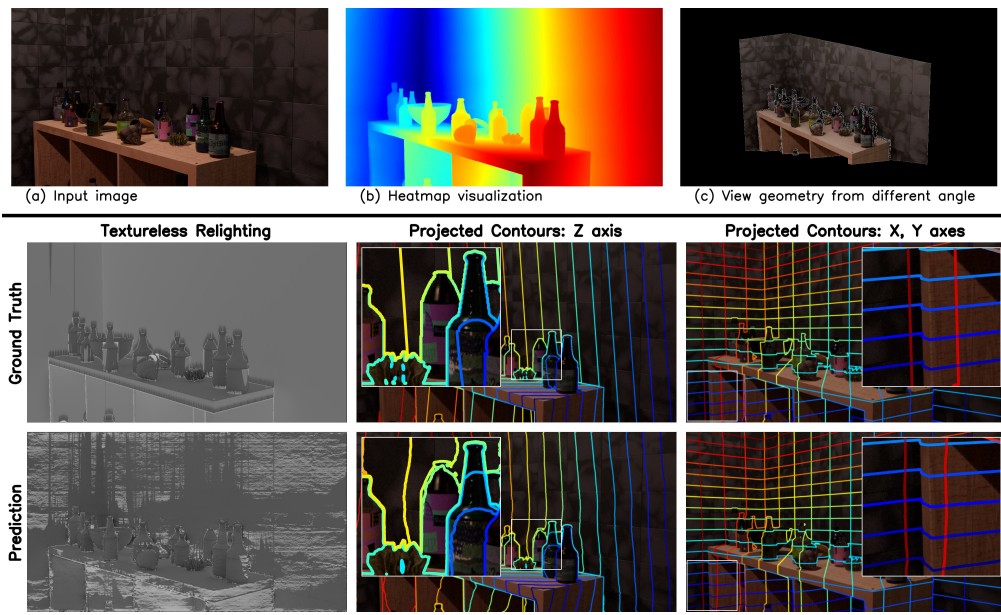

Figure 3: Top: input image and visualization of a depth using existing tools. Defects like a wavy wall are not apparent due to interference of texture. Bottom: Textureless Relighting and Projected Contours make geometric defects more apparent. Bumps on the wall are obvious.

To measure human sensitivity, we need to model human judgment as a smooth non-negative function $H(x)$ of the intensity $x$ of a perturbation. This can be tricky because it can be difficult for humans to assign consistent and well-calibrated numerical values, e.g., rate the depth map from 1 to 5.

Fortunately, we only care about the behavior near zero, which admits a simple solution. We show an RGB image and a depth map to a human annotator for a binary response: whether the depth map is the ground truth. The depth map is perturbed with varying intensities—we probe when a perturbation starts to become noticeable, which corresponds to a response of 1. we collect data points over multiple human annotators and scenes and fit a quadratic curve to obtain the derivative at zero.

Note that unlike computer metrics, humans are not given two depth maps to compare. Instead, humans are shown an RGB image and the ground truth depth map. That is, humans are comparing the ground truth depth map against their own internal 3D reconstruction. This reveals human sensitivity to perturbations in the specific context of monocular depth estimation.

## 4.1 VISUALIZATION TOOLS

In measuring human sensitivity, one issue we encounter is that the lack of effective depth visualization tools. Widely used visualizations include (1) a heatmap and (2) textured 3D point cloud viewed in a few new angles. Though these visualizations can illustrate coarse shape and severe deformations, they can mask certain defects that would be otherwise apparent. While heatmaps are widely considered insufficient, the limitation of textured point clouds is perhaps less well understood. The main issue with textured point clouds is that to the human eye, textures can create illusions of geometry that differs from the actual one, in the same way videos games use textures maps to fake geometrical details on flat surfaces. For example, Fig. 3 (b),(c) show these visualizations of one depth prediction for Fig. 3 (a). Is the wall flat? From only these two visualizations, it appears flat but is in fact wavy. To help human annotators more efficiently and effectively inspect the reconstructed 3D shape, we introduce two new visualization tools:

**Textureless Relighting.** When casting light to a geometry, different shape results in different patterns of shadow. Motivated by this, we develop Textureless Relighting by casting directional light to depth induced mesh. To prevent texture misleading users, the mesh is made textureless. Moreover, relighting under one direction of light is not enough, as it is hard to tell the shape of pixels in shadow. So the mesh is rendered under various lighting directions. Fig. 3 Bottom Col. 1 shows an example, where bumps are easily noticeable under textureless relighting.

| | Sensitivity | | | | | | | | Similarity to Human Sensitivity | SAWA-H weights | |
|---|---|---|---|---|---|---|---|---|---|---|---|
| | Surf. Ori. | Cam. Intr. | Rel. Sc. | Curv., high freq. | Curv., low freq. | Af. Depth | Af. Disparity | Boundary | | rescaled metric units | original metric units |
| AbsRel-Disparity Af. | 0.29 | 0.62 | 0.05 | 0.00 | 0.00 | 2.74 | 0.00 | 0.00 | 0.46 | 0.00 | 0.00 |
| $\delta^1$-Depth Af. (Lst. Sq.) | 0.65 | 0.37 | 0.00 | 0.00 | 0.00 | 0.00 | 2.73 | 0.04 | 0.47 | 0.00 | 0.00 |
| RMSE(log)-Depth Af. (Lst. Sq.) | 0.63 | 0.33 | 0.08 | 0.00 | 0.00 | 0.00 | 2.74 | 0.02 | 0.47 | 0.00 | 0.00 |
| $\delta^1$-Depth Af. | 0.64 | 0.46 | 0.20 | 0.00 | 0.00 | 0.00 | 2.71 | 0.06 | 0.51 | 0.00 | 0.00 |
| AbsRel-Depth Af. (Lst. Sq.) | 0.67 | 0.38 | 0.29 | 0.00 | 0.01 | 0.00 | 2.71 | 0.05 | 0.51 | 0.00 | 0.00 |
| RMSE-Depth Af. (Lst. Sq.) | 0.73 | 0.38 | 0.12 | 0.00 | 0.01 | 0.00 | 2.70 | 0.17 | 0.51 | 0.00 | 0.00 |
| RMSE-Depth Af. | 0.83 | 0.44 | 0.16 | 0.00 | 0.01 | 0.00 | 2.66 | 0.13 | 0.53 | 0.00 | 0.00 |
| $\delta^1$-Disparity Af. | 0.71 | 0.59 | 0.24 | 0.00 | 0.00 | 2.66 | 0.00 | 0.04 | 0.53 | 0.00 | 0.00 |
| RMSE-Disparity Af. | 0.61 | 1.04 | 0.03 | 0.00 | 0.00 | 2.56 | 0.00 | 0.00 | 0.53 | 0.00 | 0.00 |
| RMSE(log)-Depth Af. | 0.80 | 0.76 | 0.58 | 0.01 | 0.02 | 0.00 | 2.54 | 0.11 | 0.60 | 0.00 | 0.00 |
| RMSE(log)-Disparity Af. | 0.92 | 1.02 | 0.45 | 0.00 | 0.01 | 2.43 | 0.00 | 0.06 | 0.61 | 0.00 | 0.00 |
| WKDR-No Align. | 2.35 | 1.14 | 1.07 | 0.05 | 0.17 | 0.00 | 0.00 | 0.11 | 0.61 | 0.19 | 3.65 |
| AbsRel-Depth Af. | 1.21 | 0.79 | 0.77 | 0.01 | 0.03 | 0.00 | 2.31 | 0.06 | 0.65 | 0.00 | 0.00 |
| $\delta^1$-Depth Sc. | 0.54 | 0.50 | 0.55 | 0.00 | 0.00 | 1.90 | 1.89 | 0.02 | 0.67 | 0.00 | 0.00 |
| RMSE(log)-Depth Sc. | 0.65 | 0.50 | 0.45 | 0.00 | 0.01 | 1.66 | 2.09 | 0.05 | 0.68 | 0.00 | 0.00 |
| RMSE(log, scale-inv.)-No Align. | 0.69 | 0.55 | 0.45 | 0.00 | 0.01 | 1.58 | 2.13 | 0.06 | 0.68 | 0.00 | 0.00 |
| AbsRel$_p$-Pnt Af. | 0.61 | 0.69 | 0.41 | 0.00 | 0.01 | 1.82 | 1.92 | 0.02 | 0.69 | 0.00 | 0.00 |
| RMSE-Depth Sc. | 0.85 | 0.55 | 0.41 | 0.00 | 0.00 | 1.59 | 2.07 | 0.09 | 0.69 | 0.00 | 0.00 |
| AbsRel-Depth Sc. | 0.78 | 0.65 | 0.48 | 0.00 | 0.01 | 1.78 | 1.90 | 0.02 | 0.70 | 0.00 | 0.00 |
| $\delta^{0.125}$-Depth Af. | 1.63 | 1.06 | 1.23 | 0.00 | 0.00 | 0.00 | 1.64 | 0.06 | 0.70 | 0.00 | 0.00 |
| $\delta^{0.125}$-Disparity Af. | 1.53 | 1.05 | 1.48 | 0.00 | 0.00 | 1.54 | 0.00 | 0.05 | 0.71 | 0.14 | 0.18 |
| $\delta^1$-No Align. | 0.48 | 0.52 | 1.20 | 0.00 | 0.00 | 1.66 | 1.82 | 0.01 | 0.71 | 0.00 | 0.00 |
| RMSE(log)-No Align. | 0.71 | 0.58 | 0.76 | 0.00 | 0.01 | 1.52 | 2.07 | 0.05 | 0.71 | 0.00 | 0.00 |
| RMSE-No Align. | 1.18 | 0.53 | 0.50 | 0.00 | 0.00 | 1.38 | 2.04 | 0.09 | 0.72 | 0.00 | 0.00 |
| $\delta^{0.125}$-Depth Af. (Lst. Sq.) | 1.59 | 1.07 | 1.33 | 0.00 | 0.00 | 0.00 | 1.59 | 0.15 | 0.72 | 0.01 | 0.01 |
| AbsRel-No Align. | 0.72 | 0.57 | 0.83 | 0.00 | 0.01 | 1.87 | 1.73 | 0.02 | 0.72 | 0.00 | 0.00 |
| AbsRel$_p$-Pnt sc. | 0.73 | 1.19 | 0.44 | 0.00 | 0.01 | 1.65 | 1.77 | 0.02 | 0.73 | 0.00 | 0.00 |
| $\delta^{0.125}$-Depth Sc. | 1.26 | 1.01 | 0.69 | 0.00 | 0.00 | 1.49 | 1.64 | 0.03 | 0.77 | 0.00 | 0.00 |
| $\delta^{0.125}$-No Align. | 1.21 | 0.93 | 1.25 | 0.00 | 0.00 | 1.35 | 1.51 | 0.03 | 0.78 | 0.00 | 0.00 |
| Boundary F1-No Align. | 0.58 | 0.63 | 0.61 | 0.13 | 0.20 | 1.41 | 1.97 | 0.99 | 0.81 | 0.19 | 0.20 |
| New Metric: RelNormal | 0.52 | 0.55 | 0.03 | 1.70 | 1.28 | 1.03 | 1.21 | 0.61 | 0.87 | 0.48 | 1.94 |
| New Metric: SAWA-H (w/o RelNormal) | 1.40 | 0.96 | 0.95 | 0.12 | 0.21 | 1.16 | 1.47 | 0.79 | 0.88 | NA | NA |
| New Metric: SAWA-H | 1.27 | 0.93 | 0.68 | 1.05 | 0.85 | 1.21 | 1.18 | 0.63 | 0.97 | NA | NA |
| Human | 1.00 | 1.00 | 1.00 | 1.00 | 1.00 | 1.00 | 1.00 | 1.00 | 1.00 | NA | NA |

Figure 4: **Sensitivity:** Each row in red is a vector of exchange rates between a metric and human judgment, reflecting the sensitivity of a metric to different perturbations, using human judgment as a reference. Within a vector, a higher value means higher sensitivity of the metric to a perturbation, relative to humans. Human judgment (last row) against itself results in a vector of all 1s. Each vector can be arbitrarily scaled, equivalent to changing the unit of the metric; here the unit of the metric is rescaled such that the L2 norm of the vector is the same as a vector of all 1s, to facilitate comparisons across rows. All rows except last four are existing metrics. **Similarity to human sensitivity:** cosine similarity between each sensitivity vector and the human sensitivity vector (all 1s). **SAWA-H Weights:** SAWA-H is a new, composite metric that is a weighted average of base metrics, weights optimized to aligned with human sensitivity. RelNormal is a new base metric we introduce to allow better human alignment. The first column lists weights for averaging metrics under rescaled units, reflecting relative contributions invariant to the original choices of metric units; the second column lists equivalent weights for averaging metrics under their original units.

**Projected Contours.** This visualization is made by projecting contour lines to the 3D geometry unprojected from depth. Contour lines are projected along direction of X,Y,Z axes. Shape of contour lines can reveal the curvature of surfaces. For example, in Fig. 3 Bottom Col. 2,3, contour lines on flat planes should be straight (first row) and wavy on bumpy surface (second row).

## 4.2 Sensitivity of Existing Metrics Relative to Human Judgment

**Metrics:** We compare the sensitivity of 9 widely existing metrics against human judgment: AbsRel, AbsRel$_p$, $\delta^{0.125}$, $\delta^1$, RMSE, RMSE (log), RMSE (log, scale invariant), WKDR, and boundary F1. Because it is common to align prediction with ground truth before computing metrics, we compute these metrics with no alignment, or with one of the following alignments if applicable: scale alignment of depth, affine alignment of depth (L1 or least square), affine alignment of disparity, and scale or affine alignment of point map (Ke et al., 2024; Wang et al., 2025a). More details are in Appendix B.

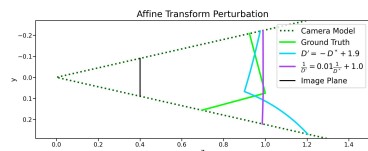

Figure 6: Side view of ground truth (green) and affine transform perturbations. Extreme affine transform parameters flatten geometry (purple) or change relative depth (blue).

Fig. 4 shows the exchange rates between every metric and human judgment under various perturbations, revealing two main findings: (1) Relative to human, existing metrics (all rows except the last four) are severely under-sensitive to curvature perturbation; (2) Humans are sensitive to affine transform of depth or disparity, which is ignored by metrics that perform the respective affine alignment. In Figure 5 and Figure 6 we use simple shapes to illustrate the theoretical causes of these observations.

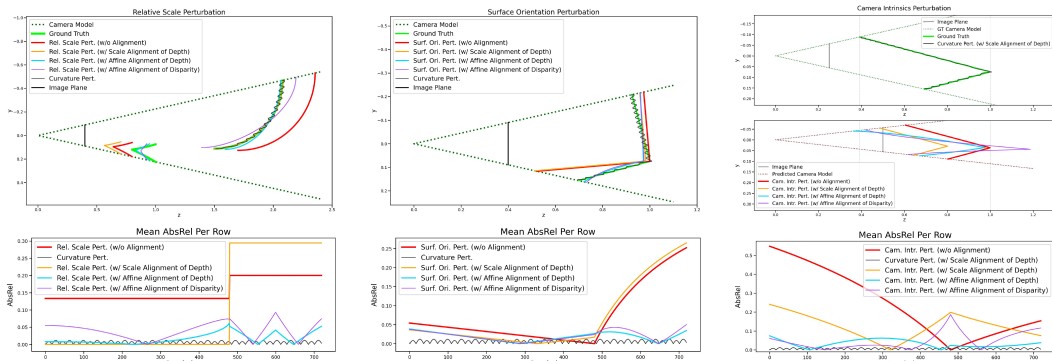

Figure 5: Top and Mid: side views of ground truth and different perturbations. Relative scale, surface orientation, and camera intrinsics perturbations (red) have good shape, but comparing with curvature perturbations (black), they have poor AbsRel, even under alignments (orange, blue, and purple). Bottom: mean AbsRel of each row. Curvature perturbation has much lower AbsRel.

## 5 SENSITIVITY ALIGNED COMPOSITION

Our sensitivity analysis of existing metrics reveals that no existing metrics align well with human judgment. How do we create a metric that does align well? A well-aligned metric can be useful for human-facing generative applications.

We introduce Sensitivity Aligned Composition (SAC), a method to compose a set of existing metrics such that the new metric achieves a desired sensitivity profile, such as that of humans or a specific application. The idea is to combine existing metrics to form a new metric, with the combination parameters optimized such that new metric is maximally similar to a target metric (which can be a black-box oracle like human judgment) in terms of sensitivity.

Given perturbations $P_1, \ldots, P_M$, we define the sensitivity vector of a metric A relative to a reference metric Z as a vector of exchange rates under the perturbations $\mathbf{R}(A; Z|\mathbf{P}) = (R(A; Z|P_1), \ldots, R(A; Z|P_M)) \in \mathbb{R}^M$. We then define the similarity of two sensitivity vectors using cosine similarity. This is because the magnitude of the sensitivity vector does not matter and is due to the choice of the unit of the metric. Let $\mathbf{T} \in \mathbb{R}^M$ be a target sensitivity vector, we can optimize the parameters $w$ of a composite metric $C(w)$ to maximize the cosine similarity between $\mathbf{R}(C(w); Z|P)$ and $\mathbf{T}$.

**Sensitivity Aligned Weighted Average (SAWA):** The simplest way to compose metrics is a non-negatively weighted average $C(w) \triangleq \sum_i w_i A_i$ of metrics $A_i$. Due to the linearity of derivative, it is easy to verify that the sensitivity vector of metric $C(w)$ relative to a reference metric $Z$ is a linear combination of the sensitivity vectors of metrics $A_i$, that is, $\mathbf{R}(C(w); Z|\mathbf{P}) = \sum_i w_i \mathbf{R}(A_i; Z|\mathbf{P})$.

Setting the reference metric to human judgment $H$ and the target sensitivity vector to $\mathbf{1}_M$ (a vector of all 1s) leads to the following optimization problem that maximizes human alignment:

$$\min_{\mathbf{w} \geq 0} \frac{\langle \sum_i w_i \mathbf{R}(A_i; H|\mathbf{P}), \mathbf{1}_M \rangle}{\| \sum_i w_i \mathbf{R}(A_i; H|\mathbf{P}) \|_2 \| \mathbf{1}_M \|_2} \tag{2}$$

This is equivalent to a convex problem which can be solved optimally and efficiently. We refer to the resulting composite metric as SAWA-H (Sensitivity Aligned Weighted Average aligned to Human judgment). Intuitively, this optimization finds the best non-negative weights to linearly combine the red rows in Fig. 4 to maximize cosine similarity with a target vector of all 1s.

**New metric: RelNormal**  Because all existing metrics are insensitive to curvature error, this limits how much we can align with human judgment. Intuitively, the rows of existing metrics in Fig. 4 cannot linearly combine to approximate well a vector of all 1s. To remedy this we introduce a new metric based on relative surface normals.

Similar to WKDR (Zoran et al. (2015)) and boundary f1 score (Bochkovskii et al. (2025)), we consider the relation between patches (sets of pixels) within the depth map. Given patches $p$ and $q$

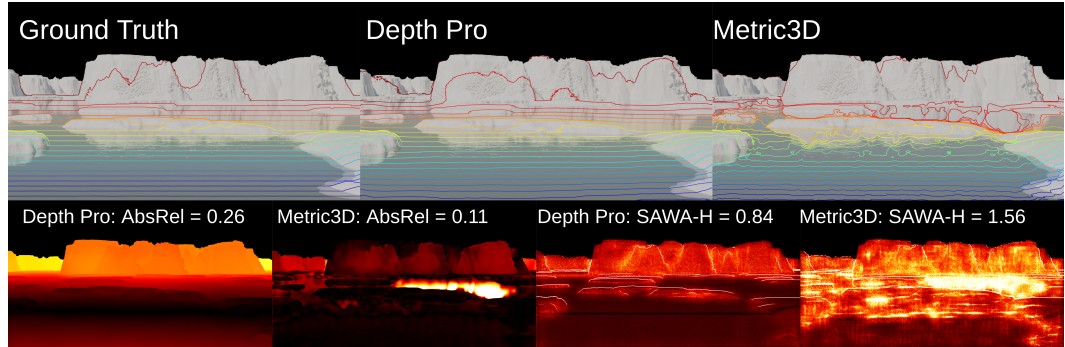

Figure 7: Disagreement between SAWA-H and AbsRel. Top row: projected contour plots of the ground truth and predicted depths. Bottom row: heatmaps for model predictions calculated with AbsRel and SAWA-H. AbsRel is computed by aligning the scale with the ground truth.

we calculate the normal of the surface described by each patch and the angle between these normals $\angle(n_p, n_q)$. We take the average over many such patches.

$$\text{RelNormal} = \sum_{p,q \in \mathcal{C}} |\angle(\hat{n}_p, \hat{n}_q) - \angle(n_p, n_q)| \tag{3}$$

If $p$ and $q$ lie on a continuous surface connected by a circular arc, this measures error in the degree of curvature. For $p$ and $q$ on separate but adjacent surfaces, this metric captures the local geometry (e.g. the angle between two walls in a room).

Our selection of patches is motivated by the following procedure: first uniformly select a pixel $I$ within the image, then select a pixel $J$ uniformly within a neighborhood of $I$, finally compute the error. To approximate this expectation over a multivariate uniform distribution we use the first $m$ elements of the Sobol sequence; this provides a deterministic algorithm for computing the metric. In the appendix we show that 1 million samples is sufficient when computing this metric on standard RGB-D datasets. We also take the average over multiple scales by downsampling the image by a factor $k$ and computing the relative normals. The following results use a neighborhood radius of 32 and scale factors $[1, 2, 4, 8]$. We compute the normals from the cross product of vectors from the top/bottom and left/right pixels adjacent to a central pixel. For datasets with noisy ground truth, it may be preferable to find a plane of best fit.

**SAWA-H (with RelNormal)** With RelNormal, we are able to achieve substantially better human alignment, improving the cosine similarity to $0.97$ with (RelNorml) from $0.88$ (without RelNormal). Fig. 4 shows the SAWA-H weights and similarity after including RelNormal.

**Examples of SAWA-H on Real Predictions** The SAWA-H metric provides different rankings among depth predictions than the AbsRel metric. In figure 7 we visualize this difference for the predictions of Depth Pro and Metric3DV2 on an Infinigen scene using error maps. This is mostly straightforward, though to compute the error map for the boundary f1 score we divide the error equally among all edge pixels where the ground truth and predicted depth map disagree. Though Metric3D achieves a smaller AbsRel error it's SAWA-H error is substantially larger. This is due to the fact that the Depth Pro model made a mistake judging the scale of the iceberg and orientation of the camera relative to ocean. Hence, the iceberg and ocean cannot both be properly aligned, causing high error. However, as seen in the projected contour plot, Depth Pro better approximates the local geometry and has a lower SAWA-H error score. We present more examples with predictions on Infinigen and iBims in the appendix (Koch et al. (2019)).

## 6  LIMITATIONS AND CONCLUSIONS

One limitation is that our list of perturbations, although representative, may not be exhaustive. Our dataset for sensitivity measurement may have room for improvement in diversity and coverage. These limitations may affect our numerical results such as particular sensitivity values. On the other hand, most of our contributions are on methodology and tools, and we expect them to stay useful and increasingly so, as more perturbations and better data become available.

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

## A  PERTURBATION ALGORITHM OF CONTROLLABLE INTENSITY

Here we provide details of algorithms to generate each perturbation.

**Relative Scale Perturbation.** We partition pixels to 3 sets, $S_{\text{near}}, S_{\text{between}}, S_{\text{far}}$. Depth of $S_{\text{near}}$ remains unchanged. Depth of $S_{\text{far}}$ is scaled by $s(s \geq 1)$, where $s$ is the parameter to control perturbation magnitude. Depth of $(i, j) \in S_{\text{between}}$ is scaled by $1 + s \cdot \frac{D^*_{i,j} - d_l}{d_r - d_l}$, where $D^*_{i,j}$ denotes ground truth depth, and $d_l$ and $d_r$ denotes depth of farthest pixel in $S_{\text{near}}$ and nearest pixel in $S_{\text{far}}$. We manually select $S_{\text{near}}$ and $S_{\text{far}}$ to be objects across occlusion boundaries, and $S_{\text{between}} = \emptyset$ for indoor scenes. For nature scenes, we first find $\text{quantile}(D^*; 0.3) \leq d_l, d_r \leq \text{quantile}(D^*; 0.7)$ with minimum number of pixels in between, and set $S_{\text{near}}, S_{\text{between}}, S_{\text{far}}$ to be pixels with depth $\leq d_l, \in [d_l, d_r]$, and $\geq d_r$.

**Surface Orientation Perturbation.** We first un-project ground truth depth to 3D, and compute ground truth surface normal following Sec. **??**. Then we rotate every ground truth surface normal by the same amount. Perturbation intensity is defined as magnitude of rotation (i.e. rotated by $s$ degrees). We further induce rotated surface normal to log depth gradient, and solve the optimization problem of minimizing L2 distance between this target log depth gradient and log depth gradient of perturbed depth.

**Camera Intrinsics Perturbation.** In this perturbation, focal length is perturbed by $s$ times, $(s \geq 1)$, i.e. $f' = sf^*$, where $f'$ and $f^*$ are predicted and ground truth focal length. $s$ controls perturbation intensity. Similar to surface orientation perturbation, we compute ground truth surface normal and further induce log depth gradient from it. But differently, here we induce log depth gradient under perturbed focal length. Again, we solve the optimization problem of minimizing L2 distance between induced log depth gradient and log depth gradient of perturbed depth.

**Curvature Perturbation.** In this perturbation, we first generate a HxW noise map $K \in \mathbb{R}^{H \times W}$, where $K_{i,j}$ are independently uniformly sampled from $[1 - s, 1 + s]$. Here $s(s \geq 0)$ controls perturbation magnitude. Then the perturbed depth, $D'$, is generated by $D' \equiv D^* \otimes \text{clip}(\text{Gaussian Smooth}(K, \sigma), \min = 0.1)$, where $D^*$ is ground truth depth, $\otimes$ denotes element-wise product, and $\sigma$ controls perturbation frequency. We choose $\sigma = 1$ for high frequency perturbation, and $\sigma = 10$ for low frequency.

**Affine Transform Perturbation.** We study affine transform of depth and disparity. For affine transform of depth, perturbed depth $D'$ follows $D' \equiv \frac{1}{s}D^* + \text{median}(D^*) - \text{median}(\frac{1}{s}D^*)$, where $s \geq 1$ controls perturbation magnitude. And for affine transform of disparity, perturbed depth $D'$ follows $\frac{1}{D'} \equiv \frac{1}{s}\frac{1}{D^*} + \text{median}(\frac{1}{D^*}) - \text{median}(\frac{1}{s}\frac{1}{D^*})$, where $s \geq 1$ controls perturbation magnitude.

**Boundary Perturbation.** We apply mean filter of patch size $2s + 1$, where $s \geq 0$ controls perturbation magnitude.

## B  METRIC DEFINITIONS

Here we provide definitions for the metrics used in figure 4. These are all well-known or variants of well-known metrics.

For a ground truth depth map $z$ and a predicted $\hat{z}$, AbsRel is defined as the average of $|z_i - \hat{z}_i|/z_i$. Following Wang et al. (2025a), we define $\text{AbsRel}_p$ on point maps $p \in \mathbb{R}^{H \times W \times 3}$ by the average of $\|p_i - \hat{p}_i\| / \|p_i\|$.

$\delta^1$ is defined as the fraction of pixels such that $\max\left\{d_i/\hat{d}_i, \hat{d}_i/d_i\right\} < 1.25$. We define a stricter metric $\delta^{0.125}$ by the fraction of pixels such that $\max\left\{d_i/\hat{d}_i, \hat{d}_i/d_i\right\} < 1.25$.

RMSE and RMSE (log) follow standard definitions. RMSE (log, scale invariant) is the defined as

$$\text{Log RMSE SI} = \sqrt{\frac{1}{n}\sum_{i=1}^{n}(\log z_i - \log \hat{z}_i + \alpha)^2)} \quad \text{where } \alpha = \frac{1}{n}\sum_i(\log \hat{z}_i - \log z_i).$$

This follows the definitions in Ke et al. (2024); Eigen et al. (2014).

Finally, WKDR follows the definition given in Yin et al. (2021) based on relative depth relationships between pairs of pixels. Boundary F1 score follows the definition in Bochkovskii et al. (2025) to compute an F1 score over the edges in the predicted and ground truth depth maps.

The affine and scale alignment procedure used throughout is identical to MoGe (Wang et al. (2025a)). Notably, affine alignment for the point cloud metric $AbsRel_p$ is not the same as affine alignment of the depth. "Lst. Sq." denotes least squares alignment on depth, as performed by Marigold (Ke et al. (2024)).

## C  RELATIVE NORMAL

### C.1  SAMPLING PROCEDURE

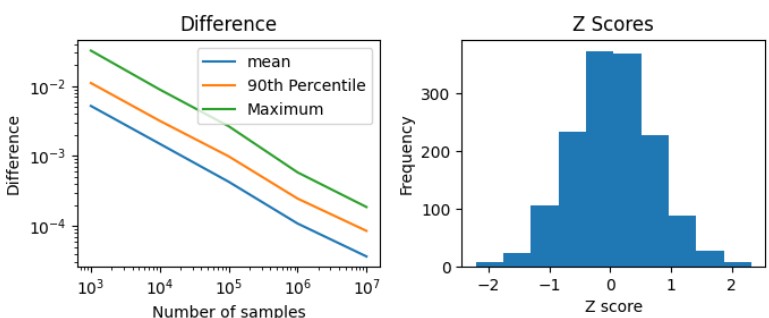

Figure 8: Comparison between deterministic and random sampling algorithm. The left plot shows the difference between the deterministic algorithm with $n$ samples and the random algorithm with $10^8$ samples. The maximum difference with 1M samples is $5.84 \times 10^{-4}$ while the mean difference is $1.08 \times 10^{-4}$. The right plot shows the z scores of the deterministic algorithm with 1M samples, compared to a distribution of the randomized algorithm with 1M samples computed with 30 different seeds.

When computing the relative depth metric, we wish to ensure that our deterministic sampling technique accurately approximates the true mean of the distribution. In figure 8 we display the difference between the metric computed with the deterministic algorithm and the metric computed with $10^8$ random samples. We perform the error computations using predictions from Depth Pro (Bochkovskii et al. (2025)), UniDepthV2 (Piccinelli et al. (2025)), MoGeV1 (Wang et al. (2025a)), Metric3DV2 (Hu et al. (2024)), and MoGeV2 (Wang et al. (2025b)) on 100 images from the iBims dataset and 100 images from the Virtual KITTI dataset Koch et al. (2019); Cabon et al. (2020). The maximum error $5.84 \times 10^{-4}$ and average error of $1.08 \times 10^{-4}$ is sufficiently small to justify using $10^6$ samples. We also analyze the $z$-score of the deterministic computation compared to random sampling. This suggests that using the Sobol sequence does not introduce unexpected irregularities.

## D  SAWA-H COMPARISON EXAMPLES

See figure 9.

## E  DATASET IMAGES SAMPLE

See figure 10.

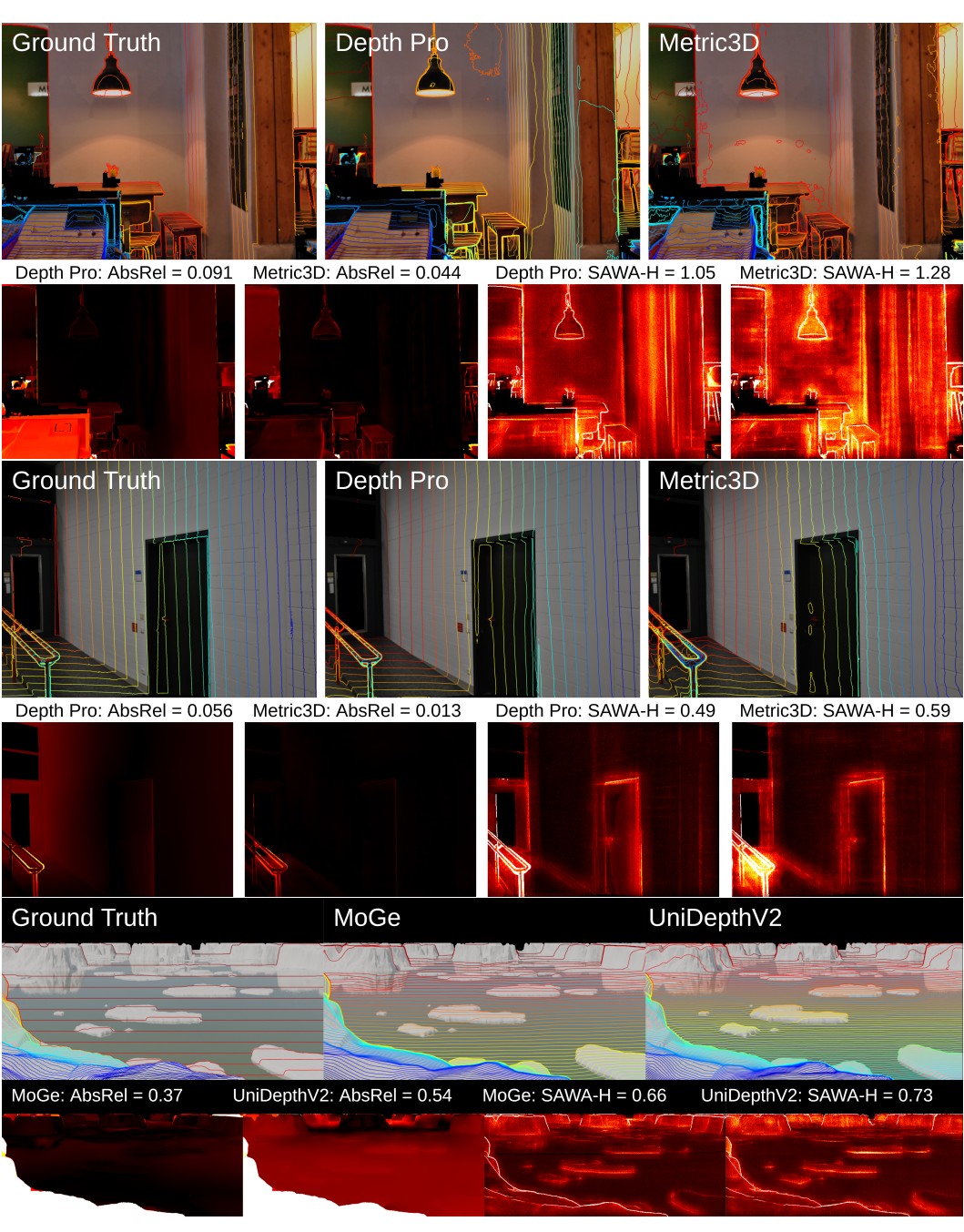

Figure 9: Additional comparisons between SAWA-H and AbsRel. The top row displays contour plots of the ground truth and predicted depths. The bottom row displays heatmaps for model predictions calculated with AbsRel and SAWA-H. AbsRel is computed by aligning the scale with the ground truth.

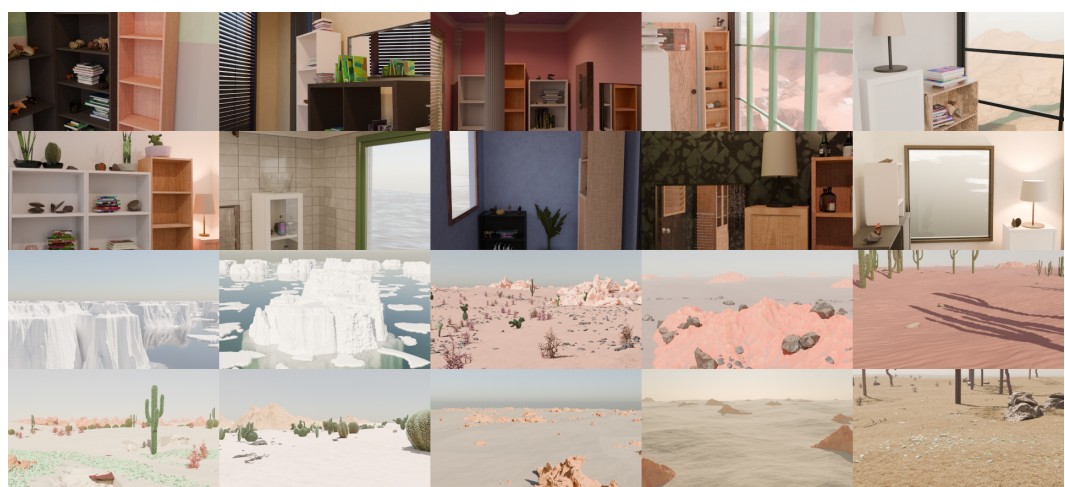

Figure 10: Sample of 20 Images from the Sensitivity Dataset

