# OpenReview forum: "How to evaluate monocular depth estimation?"
_ICLR.cc/2026/Conference — ICLR 2026 Conference Withdrawn Submission_

### Official Review · Reviewer_K12W · 2025-10-18

**Soundness:** 3
**Presentation:** 3
**Contribution:** 2
**Rating:** 6
**Confidence:** 2

**Summary:**

This paper propose to address the challenge of evaluating monocular depth estimation models, highlighting the lack of standardization and unclear trade-offs among existing metrics. The core technical idea involves quantitative sensitivity analysis of metrics by measuring their responses to 6 interpretable perturbations of ground truth depth (surface orientation, camera intrinsics, relative scale, curvature at high/low frequencies, affine transforms, and boundaries).
For each perturbation, the paper establishes an exchange rate between two metrics under this this perturbation. Then exchange rates across metrics and perturbations are compared to better capture the insights of various metrics. The paper further measures the sensitivity of human judgment, for which two novel visualization tools, i.e., Textureless Relighting and Projected Contours, are introduced to reveal geometric defects. A new metric (RelNormal) based on surface normal angles to address curvature insensitivity, and a Sensitivity Aligned Composition (SAC) method to create SAWA-H, which is a new, composite metric that aligns better with human perception. Experimental results on a 95-scene dataset show SAWA-H outperforms existing metrics in human alignment, and the qualitative examples demonstrate different rankings from AbsRel on real predictions from models like Depth Pro and Metric3DV2.

**Strengths:**

The integration of human judgment as a reference through binary annotations and novel visualizations is interesting, which provides a novel perspective for correlating vision tasks with perceptual relevance.

The use of exchange rates from perturbation derivatives enables systematic, quantitative comparison of metric behaviors

The new tools (Textureless relighting and Projected Contours) make subtle defects more apparent, as shown in examples where traditional textured point clouds will fail.

**Weaknesses:**

The core insights of the paper has been proposed and studied in many previous works. For example, it seems the idea of analyzing metric sensitivity to perturbations echoes prior work [1][2] as follows. The BenchDepth also reveals the issues of alignment-based metrics that introduce biases, favor certain depth representations.

[1] Improving Domain Generalization in Self-Supervised Monocular Depth Estimation via Stabilized Adversarial Training

[2]RoboDepth: Robust Out-of-Distribution Depth Estimation under Corruptions

The dataset uses Infinigen for synthetic scenes. Although it is claimed "We choose to use synthetic data because realworld data are limited in the diversity of scenes and availability of dense depth ground truth. " However, as the core task of this work is to benchmark the depth evaluation, real-world evaluation is very important. There are still many workarounds: using depth inpainting for LIDAR depth maps, or using a part of indoor real-world data. This is crucial to validate generalizability.

The paper claims that human sensitivity provides clues for "human-level visual capabilities", but this lacks experimental validation beyond annotations; no downstream tasks (e.g. robotics) test if SAWA-H improves performance over existing metrics.

It would be good to conduct ablation studies on the key perturbation types, such as curvature frequencies, in order to assess their individual impact on exchange rates or human annotations. Also, the visualization tools seem to be dependent on the rendering effect. Will this affect human judgment? This point is not discussed in detail.

**Questions:**

Are all six perturbation types necessary? Or if alternatives (e.g. noise patterns) yield similar insights?

Just curious: can we extend SAC beyond weighted averages to non-linear compositions (e.g. via neural networks trained on perturbation data) for capturing complex interactions?

For the conclusion part, I only see limitations and no actual summary of the paper's key content.

---

### Official Review · Reviewer_EiYb · 2025-10-26

**Soundness:** 1
**Presentation:** 2
**Contribution:** 1
**Rating:** 2
**Confidence:** 4

**Summary:**

The paper empirically compares the sensitivity of existing performance metrics for depth reconstruction to perturbations of the ground-truth. It introduces a comparison with humans and shows that existing metrics are not able to capture human sensitivity to non-linear deformations of partially planar scenes, if humans are shown an appropriate visualisation. Based on this, they generate a new metric to better capture human sensitivity.

The discussion of the human experiments is somewhat muddled. Humans are never given access to ground-truth, but instead have to determine based on a visualization if a reconstruction is implausible. As such, humans are identifying violations of their a priori assumptions, and it does not measure their sensitivity to change. All of the discussion about this, e.g., line 84: ' measure the sensitivity of human judgment to various perturbations' is misleading.

The end result, that humans notice if something that should be piecewise planar (e.g. fig 3.) isn't (given suitable visualizations) is unsurprising, but there's no real reason to think that the conclusions generalize to, e.g,. nature scenes, where there is no obvious planar prior. These limitations are not discussed.

The choice of perturbations is unmotivated and, as such, the relevance of the rest of the analysis is unclear.

**Strengths:**

The authors have correctly identified that there is a flood of possible measures for 3d reconstruction, and that it is genuinely hard to know which measures are appropriate to use.

The approach of comparing the sensitivity of some existing measures to perturbations is novel (at least in monocular reconstruction), and the core idea that one kind of useful metric would closely track human perception is convincing.

**Weaknesses:**

This is an ambitious project that falls short of what it sets out to do.

The human experiments do not measure sensitivity to perturbations.

The human experiments do not make the case that people can identify curvature errors when the target reconstruction is non-piecewise planar.

The choice of perturbations is unmotivated, and not grounded in terms of the typical errors present in reconstruction.

The comparisons of the new metric with abs rel feels extremely cherry-picked, and is limited to a few small examples.

Finally, visualization of surface normals, and evaluating their error is common in 3d reconstruction. e.g. see https://openaccess.thecvf.com/content/ICCV2021/papers/Long_Adaptive_Surface_Normal_Constraint_for_Depth_Estimation_ICCV_2021_paper.pdf

While the visualization proposed in the paper is different, there's no discussion of existing surface normal visualization, and nothing to suggest that the new approach is better. Similarly, there are many existing measures for normal error (see above paper for a small subset). However, these are not discussed, and there's nothing to suggest that the proposed cosine similarity improves on existing measures.

**Questions:**

I'm not sure that all the weaknesses listed can be addressed.

But, if you could explain how you would go about doing so, or why they are of limited concern, it would be appreciated.

---

### Official Review · Reviewer_PNeW · 2025-10-29

**Soundness:** 2
**Presentation:** 2
**Contribution:** 2
**Rating:** 2
**Confidence:** 4

**Summary:**

The paper identifies the lack of consistent evaluation metrics in monocular depth estimation and addresses this gap through a sensitivity study that measures how existing metrics respond to different types of perturbations. The work is primarily experimental, introducing the concept of an “exchange rate” to quantify metric sensitivity. Observing that existing metrics are poorly aligned with human judgment, especially insensitive to curvature perturbations, the authors propose a new surface-normal-based metric, RelNormal, and a composite metric, SAWA-H, which combines existing metrics to better match human perceptual preferences.

**Strengths:**

The paper identifies that the evaluation metrics for monocular depth estimation remains unsettled and are far from ideal. This recognition is valuable and addresses a foundational issue that is often overlooked.

The finding that existing metrics are largely insensitive to curvature perturbations is concrete. While practitioners have informally noted that current metrics fail to penalize geometric distortions such as wavy surfaces, this paper provides a systematic empirical confirmation.

The proposed sensitivity study introduces an interpretable way to probe how metrics respond to controlled perturbations in geometry, scale, or camera intrinsics. This approach quantifies the responsiveness of each metric to meaningful physical variations. Even if the framework is somewhat heuristic, it represents a creative and potentially reusable idea for diagnosing and understanding metric behavior.

The implementation of custom visualization tools such as “Textureless Relighting” and “Projected Contours” demonstrates notable technical effort.

**Weaknesses:**

After reading the paper, several major concerns arise regarding its scope, technical contributions, the design of the human study, and its connection to model performance.

1. Scope and framing. The title and framing suggest a comprehensive solution to the problem of “how to evaluate monocular depth estimation,” yet the paper focuses on a narrow set of perturbations and ultimately proposes a single human-aligned metric (SAWA-H). The scale of the contribution does not justify such an ambitious framing. The paper also makes multiple bold but shallow claims, for example, stating that “how to evaluate remains an open problem” and that “we introduce a principled method”, which are overstated given that well-established metrics such as AbsRel, SqRel, RMSE, log RMSE, and threshold-based accuracy are already widely adopted and understood within the community. While these metrics are not perfect, the paper’s claims are overly broad, and the title comes across as somewhat sensational rather than precise, i.e., it reads like clickbait.


2. Technical contribution. It is not surprising that existing metrics are under-sensitive to perturbations involving surface normals. While incorporating curvature sensitivity is a valuable direction, the paper fails to address the key challenges it highlights in the introduction, such as scale ambiguity, unbounded depth, and unknown intrinsics (L041-L058). This again reflects a mismatch between the paper’s stated scope and its actual contributions. Despite claiming to support “downstream applications” (L56–58), the work provides no experiments or analysis demonstrating that using SAWA-H as a metric reflects performance in any downstream task. The paper should clarify what these downstream applications are and present corresponding evidence, and what the technical novelty of the paper is.


3. Human study design and motivation. The motivation for aligning evaluation metrics with human perception is underdeveloped, and there is no evidence that human-aligned metrics lead to better downstream outcomes. Furthermore, the human subject study is poorly described: key details such as the number of participants, their expertise, the recruitment procedure, and measures of annotation consistency are missing. This lack of methodological rigor undermines the credibility of the human sensitivity analysis, making the results appear arbitrary and scientifically weak. I suggest that the authors review standard human-subject study protocols in both computer vision and psychophysics and ensure that future studies include clear participant details, procedures, and statistical validation.

There are also minor weaknesses, including

4. The proposed metric is not tested on real-world monocular depth data, such as KITTI and NYU-V2. While the authors should not be required to test their method on an excessive amount of additional data, missing any results or discussion on these two most commonly adopted datasets is questionable.

5. Introducing "exchange rate" seems arbitrary. It feels more heuristic than a really principled way to measure sensitivity.

**Questions:**

1. Human Subject Study: How many participants were recruited? What is the recruiting criteria and were they domain experts or from the general public? How were consistency and bias measured (any statistical validation)?

2. Why should human perceptual sensitivity define the “ideal” evaluation metric for depth estimation?

3. Is there evidence that SAWA-H correlates better with performance on any downstream tasks (e.g., SLAM, new view synthesis, navigation, etc.)?

4. Beyond confirming that curvature matters, what novel understanding of monocular depth evaluation does this paper add?

5. Does the proposed metric generalize beyond synthetic data to real-world datasets like NYU-V2 or KITTI?

---

### Official Review · Reviewer_nY48 · 2025-11-01

**Soundness:** 4
**Presentation:** 3
**Contribution:** 3
**Rating:** 8
**Confidence:** 4

**Summary:**

This paper argues that current evaluation metrics (e.g., AbsRel, RMSE, δ thresholds) for monocular depth estimation fail to capture human-perceived depth quality. The authors propose evaluating MDE models through sensitivity, defined as the extent to which small, systematic perturbations in depth maps lead to detectable differences for downstream perception. They introduce controlled perturbation procedures and a human perceptual study to validate whether sensitivity correlates with human judgments. Experiments across several depth models and datasets show that traditional metrics do not reliably predict perceived depth quality, whereas sensitivity aligns more closely with human assessments.

**Strengths:**

1. The paper clearly explains why conventional depth metrics may not reflect how humans perceive depth quality and why this matters for applications like AR/VR, 3D editing, and content creation. The definition of core concepts such as sensitivity is intuitive, operationalized precisely, and the perturbation process is described in a reproducible manner.
2. The design of perturbation families (e.g., structural, edge-based, region-based distortions) appears systematic and justified.
3. The human evaluation is well-motivated and improves credibility. It supports the central claim that human perception deviates from what common depth metrics measure. The proposed visualization tools also makes sense intuitively.

**Weaknesses:**

1. For the human study, the user population size and demographics are not extensively discussed. If users have prior familiarity with depth perception tasks (or are non-experts), this could influence results. A breakdown or justification would strengthen confidence.
2. The paper claims that  SAWA-H better reflects “real-world usefulness,” but the benchmark focuses on perceptual assessment. It would be valuable to show whether  SAWA-H predicts performance in at least one downstream application, even if only exploratory.
3. Minor formatting issues exist, see line 607

**Questions:**

1. Can you provide a more comprehensive description of the design for human subject study? The current paper lacks detail on this experiment.
2. Are there any correlations between the proposed SAWA-H and dowstream tasks?
3 Is the sensitivity score sensitive to image content? (e.g., indoor vs. outdoor, clutter vs. low-texture)

---

### Note · Authors · 2025-11-12

I have read and agree with the venue's withdrawal policy on behalf of myself and my co-authors.